# Coupled Mode Design of Low-Loss Electromechanical Phase Shifters

**Nathnael S. Abebe \***, **Sunil Pai**, **Rebecca L. Hwang**, **Payton Broaddus**, **Yu Miao** and **Olav Solgaard \***

Ginzton Laboratory, Stanford University, Stanford, CA 94305, USA; sunilpai@alumni.stanford.edu (S.P.); rebeccahwang3@gmail.com (R.L.H.); broaddus@stanford.edu (P.B.); miaoy11@alumni.stanford.edu (Y.M.)
* Correspondence: nsabebe@stanford.edu (N.S.A.); solgaard@stanford.edu (O.S.);
Tel.: +1-650-224-7177 (N.S.A. & O.S.)

**Abstract:** Micro-electromechanical systems (MEMS) have the potential to provide low-power phase shifting in silicon photonics, but techniques for designing low-loss devices are necessary for adoption of the technology. Based on coupled mode theory (CMT), we derive analytical expressions relating the loss and, in particular, the phase-dependent loss, to the geometry of the MEMS phase shifters. The analytical model explains the loss mechanisms of MEMS phase shifters and enables simple optimization procedures. Based on that insight, we propose phase shifter geometries that minimize coupling power out of the waveguide. Minimization of the loss is based on mode orthogonality of a waveguide and phase shifter modes. We numerically model such geometries for a silicon nitride MEMS phase shifter over a silicon nitride waveguide, predicting less than $-0.08$ dB loss over a $2\pi$ range and $-0.026$ dB loss when optimized for a $\pi$ range. We demonstrate this design framework with a custom silicon nitride process and achieve $-0.48$ dB insertion loss and less than 0.05 dB transmission variation over a $\pi$ phase shift. Our work demonstrates the strength of the coupled mode approach for the design and optimization of MEMS phase shifters.

**Keywords:** silicon photonics; integrated optics; MEMS; phase shifters

## 1. Introduction

Photonics integrated circuits (PICs) have shown significant improvements in architecture, electronic integration, and device performance [1–3]. Given their rapid development, these PICs have found a host of applications beyond telecommunications, including classical and quantum computing [4], sensor arrays [5], and light detection and ranging (LIDAR) [6]. Common to most system-level photonics designs is optical phase/path length tuning with thermo-optic phase shifters [7,8]. Unfortunately, all thermo-optic phase shifters are dissipative, exhibiting large static power consumption and strong cross-talk, leading to the need for better phase shifter technologies.

Alternatives to the thermo-optic phase shift include the plasma effect [9], electro-optic effects [10], phase change [11], and micro-electromechanical systems (MEMS) [12,13]. MEMS phase shifters are typically configured with a perturbing element that interacts with the evanescent field of the waveguide. By placing the perturbing element in different positions relative to the waveguide, the propagation constant of the waveguide mode is changed, leading to a phase shift for the interaction length. Positioning of the phase shifting element can be achieved through any micromechanical actuation mechanism [12]. Electrostatic actuation is the most common mechanism because it leads to small switching energies and negligible static power consumption. As a switching technology, MEMS have shown great performance with low optical losses [14–19], but there are fewer demonstrations of MEMS phase shifters with low optical loss [20–24] because of a phase-dependent loss caused by optical coupling in the phase shifting block. This paper details a coupled mode analysis of MEMS phase shifters. The theory clarifies the loss mechanisms, pinpoints the sources of phase-dependent loss, provides design guidelines for low loss, and allows for

analytical verification of proposed designs. To demonstrate the practicality of our analysis, we design, simulate, and test a low-loss MEMS phase shifter in a custom silicon nitride photonics stack.

## 2. Coupled Mode Analysis

The mode of operation of MEMS phase shifters is typically to introduce or remove an element in a waveguide system to change the propagation constant of the guide [12]. The shape and material of the perturbing element is crucial to the optical performance, so a method of analyzing those parameters is of practical importance. As shown in Figure 1a, we define a system of two arbitrary guides A and B that represent the original waveguide and the phase shifter, respectively. If we assume the real modes of the system can be well-represented as a linear superposition of the guided modes of guides A and B, we can apply coupled mode theory (CMT) to estimate the phase shift per unit length and transmission of the system [25].

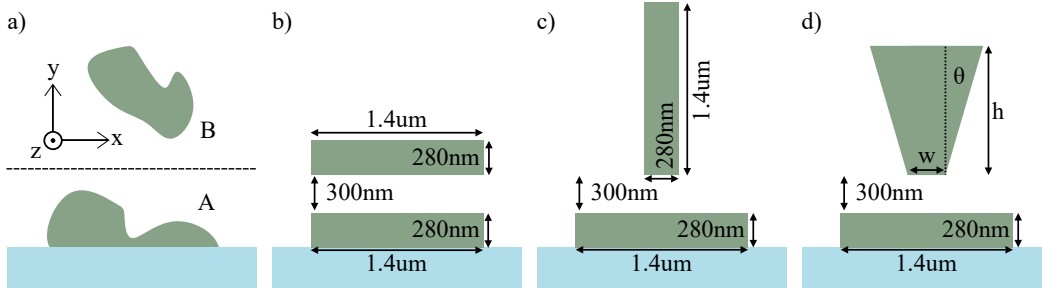

**Figure 1.** (**a**) An abstraction of a MEMS phase shifter as two guides A and B. (**b**) A TE-like phase shifter with identical dimensions to the single mode waveguide it affects. (**c**) A TM-like phase shifter with the same dimensions as the waveguide, but rotated 90°. (**d**) A phase shifter parameterized by height, base width, and sidewall angle.

Following the multi-waveguide formalism presented in [26], we define the full coupled mode system consisting of $(n + m)$ modes where $\mathbf{a}(z)$ is a vector of $n$ modes from the guide A, and $\mathbf{b}(z)$ is a vector of $m$ modes from guide B.

$$\frac{d}{dz}\begin{bmatrix} \mathbf{a}(z) \\ \mathbf{b}(z) \end{bmatrix} = j(\mathbf{B} + \bar{\mathbf{C}}^{-1}\mathbf{K})\begin{bmatrix} \mathbf{a}(z) \\ \mathbf{b}(z) \end{bmatrix} \tag{1}$$

An exact formalism can include the continuum of radiation modes as mentioned in the work of Hardy and Streifer [27], but we restrict the analysis to only the guided modes, as that captures the majority of the energy transfer. The evolution matrix $(\mathbf{B} + \bar{\mathbf{C}}^{-1}\mathbf{K})$ in Equation (1) consists of a diagonal matrix, $\mathbf{B}$, of propagation constants; a square matrix, $\bar{\mathbf{C}}$, of interaction strength integrals; and a square matrix, $\mathbf{K}$, of induced polarization integrals. The elements of the $\mathbf{K}$ and $\bar{\mathbf{C}}$ matrices are $\tilde{K}_{b_j a_k}$ and $(C_{b_j a_k} + C_{a_k b_j})/2$, respectively.

$$\tilde{K}_{b_j a_k} = \frac{\omega}{4} \iint_{-\infty}^{+\infty} \Delta\epsilon^{(a_k)}(\mathbf{E}_t^{(b_j)} \cdot \mathbf{E}_t^{(a_k)} - \frac{\epsilon^{(b_j)}}{\epsilon} E_z^{(b_j)} E_z^{(a_k)})dxdy \tag{2a}$$

$$C_{b_j a_k} = \frac{1}{2} \iint_{-\infty}^{+\infty} (\mathbf{E}_t^{(a_k)} \times \mathbf{H}_t^{(b_j)}) \cdot \hat{z} \, dxdy \tag{2b}$$

The terms $\tilde{K}_{b_j a_k}$ and $C_{b_j a_k}$ are overlap integrals that describe induced polarization and interaction strength, respectively. We calculate the integrals at a fixed frequency $\omega$, and we define the full permittivity distribution as $\epsilon$ and the individual guide distribution as $\epsilon^{(i)}$. The induced polarization and interaction strength interactions depend strongly on the transverse electric fields $\mathbf{E}_t^{(i)} = (E_x^{(i)}, E_y^{(i)})$ and perturbation $\Delta\epsilon^{(i)}$. The perturbation to waveguide A is the material that constitutes waveguide B, and the perturbation to waveg-

uide B is the material that constitutes waveguide A. The longitudinal electric field, $E_z^{(i)}$, is typically orders of magnitude smaller than the transverse fields, so it can be negligible.

The general solution to Equation (1) is given by the projection of the starting vector of modes to a set of super-modes whose phases evolve over the propagation distance z and are then projected back onto the original vector of modes.

$$\begin{bmatrix} \mathbf{a}(z) \\ \mathbf{b}(z) \end{bmatrix} = \mathbf{A} \exp(j\mathbf{\Gamma}z)\mathbf{A}^{-1} \begin{bmatrix} \mathbf{a}(0) \\ \mathbf{b}(0) \end{bmatrix} \tag{3}$$

In Equation (3), the matrix $\mathbf{A}$ has columns that are the eigenvectors of $(\mathbf{B} + \bar{\mathbf{C}}^{-1}\mathbf{K})$. These eigenvectors describe the mix of individual modes that compose the super-modes, such that $\mathbf{A}^{-1}$ is the projection operation from the individual modes to the super-modes. The super-modes themselves evolve according to $\exp(j\mathbf{\Gamma}z)$, where $\mathbf{\Gamma}$ is a diagonal matrix of the super-mode propagation constants. Considering this general solution, a potential design methodology to achieve low loss is to construct a particular projection operation. Starting with a single mode of interest, we can design the coupled system to project that mode onto a single super-mode and then back after a distance $z$. This methodology lends itself well to numerical methods that we can frame into an optimization problem later.

If we limit the number of modes to one for guide A and one for guide B, we can use the familiar coupled mode solutions to the $2 \times 2$ evolution equation. This form gives insight into how to bound the loss based on the off-diagonal elements of $\mathbf{K}$, which are the coupling constants of the system. We define the coupled mode system consisting of the perturbed fundamental modes as follows:

$$\frac{d}{dz}\begin{bmatrix} a(z) \\ b(z) \end{bmatrix} = j \begin{bmatrix} \gamma_a & \kappa_{ab} \\ \kappa_{ba} & \gamma_b \end{bmatrix} \begin{bmatrix} a(z) \\ b(z) \end{bmatrix}. \tag{4}$$

The propagation constants $\gamma_a$ and $\gamma_b$ relate the phase evolution of the perturbed guided modes of $A$ and $B$ to their respective modal amplitudes $a(z)$ and $b(z)$. The coupling constant $\kappa_{ba}$ relates the evolution of $b(z)$ to the modal amplitude $a(z)$, and vice versa for $\kappa_{ab}$. These constants can be derived from the polarization integrals of Equation (2a,b) using the following forms.

$$\gamma_a = \beta_a + \frac{\tilde{K}_{aa} - \tilde{K}_{ba}\bar{C}}{1 - \bar{C}^2} \tag{5a}$$

$$\kappa_{ba} = \frac{\tilde{K}_{ba} - \tilde{K}_{aa}\bar{C}}{1 - \bar{C}^2} \tag{5b}$$

$$\bar{C} = \frac{C_{ab} + C_{ba}}{2} \tag{6}$$

The exact relation of the propagation and coupling constants is derived from the Lorentz reciprocity of this lossless system [25].

$$(\kappa_{ba} - \kappa_{ab}) = \delta(C_{ba} + C_{ab}) \tag{7a}$$

$$\delta = \frac{\gamma_b - \gamma_a}{2} \tag{7b}$$

The constants $C_{ba}$ and $C_{ab}$ are the modal overlap integrals that describe the strength of the interaction between guides $A$ and $B$. Usually for coupled waveguides, we assume weak interactions of the individual guides, $C_{ba} \to 0$ and $C_{ab} \to 0$, such that $\kappa_{ba} = \kappa_{ab} = \kappa$. In our analysis, we cannot make the standard simplifications because the interaction strength is large and the detuning, $\delta$, of the propagation constants of the asymmetric guides does not tend to zero in general. Under these conditions, the maximum fractional power coupled

out of guide A is $\frac{\kappa_{ba}\kappa_{ab}}{\kappa_{ba}\kappa_{ab}+\delta^2}$. To minimize coupling loss in guide A, we should minimize the coupling constants, $\kappa_{ba}$ and $\kappa_{ab}$, or maximize the detuning $\delta$.

The induced polarization integrals, as defined in Equation (2a), are the dominant terms in the coupling constants that drive the power exchange. The vector nature of the fields in Equation (2a) is a key consideration in reducing this power exchange. To proceed, we impose practical constraints and analyze three geometries for guide B while fixing waveguide A to be a rectangular cross-section that supports a single TE-like mode, which is the situation in most photonic integrated circuits.

The first geometry for guide B has a rectangular cross-section of similar dimensions to waveguide A, as shown in Figure 1b. If chosen to be exactly like A, the familiar directional coupler is formed. In this case, the coupling constant is large, and the two fundamental modes exchange power periodically. The effective phase shift from changing the separation of the guides would be large, as would the lost power from waveguide A.

Figure 1c shows a perturbation that carries a fundamental mode with parity, or polarization, orthogonal to waveguide A. Here, guide B is identical to guide A, but rotated 90° about the direction of propagation. The transverse fields of the mode of guide B would be identical to guide A, but with the $E_x$ and $E_y$ fields exchanged. If we substitute this mapping, $E_x^{(b)}(x-x_b, y-y_b) = -E_y^{(a)}(x-x_a, y-y_a)$ and $E_y^{(b)}(x-x_b, y-y_b) = E_x^{(a)}(x-x_a, y-y_a)$, into Equation (2a) and set $x_b, y_b = x_a, y_a$, we see that $\tilde{K}_{ba}$ is minimized with respect to the transverse fields and only scales with the negligible longitudinal fields. If guide A has a symmetry axis across the vertical line $x = x_a$ and supports a TE-like mode, the rotated guide B would be directly overhead and support a TM-like mode. The simple, intuitive, and powerful design rule that follows is that the mode of the waveguide and the phase shifter should be orthogonal!

For simulation purposes, we introduce a third geometry for guide B, that of a symmetric trapezoid parameterized by the base width, height, and sidewall angle shown in Figure 1d. These three parameters can be chosen to recreate the first two geometries. The trapezoidal structure is a generalization of shapes that often results in practical microfabrication.

## 3. Optimization

Using the three parameters of base width, height, and sidewall angle, we optimize the trapezoidal geometry and show how CMT insights simplify the optimization. In our analysis, we chose the length of the interaction region, $l_{int}$, as the critical dimension that constrains the size. We continue to assume that the phase shifting structure has translational symmetry in the direction of propagation, so that the change in phase shift, $\Delta\phi$, is equal to the product of the length of the interaction region and $\Delta\gamma_{a0}$, where $\Delta\gamma_{a0}$ is the change in propagation constant of the perturbed fundamental mode in waveguide A. This leads to the form of the minimum phase shift, $\Delta\phi_{min}$, constraint

$$l_{int}\Delta\gamma_{a0} \geq \Delta\phi_{min}. \tag{8}$$

The change in the propagation constant, $\Delta\gamma_{a0}$, for the MEMS phase shifter is calculated as the difference in the propagation constant, $\gamma_{a0}$, when the perturbation B is in the initial and final positions.

A straightforward quantity to maximize would be $T_{min}$, the minimum transmission of the MEMS phase shifter across the operational range. This quantity can be most accurately calculated from a full 3D finite-difference time-domain (FDTD) simulation, but capturing the minimum in the operational range would require sampling with many FDTD simulations for each design iteration. This computation quickly becomes prohibitively expensive for an optimization. Instead, it would be advantageous to use the insights from CMT to define optimization problems that require less simulation time. The evolution Equation (1) and the general solution Equation (3) of the higher-order coupled mode system lead us to two optimization formulations for the trapezoidal geometry.

The first optimization is based on the coupling terms in Equation (1). The two-mode CMT shows that the maximum loss for a coupling interaction is bound by the coupling constants. Further, Equation (2a,b) show that the coupling strength increases as the two modes are brought closer together. For the first order, we assume that the pairwise coupling to the fundamental mode of waveguide A dominates the loss for the higher-order coupled mode system. Consequently, the objective is to minimize the coupling constants, measured at the minimum separation in the MEMS phase shifter geometry, between the fundamental mode of waveguide A to the remaining $(n + m - 1)$ modes, such that (s.t.) the phase shift is greater than the required minimum phase shift, $\Delta\phi min$. At the minimum separation, the optimization is as follows:

$$\min_{w,h,\theta} \sum_{j=1}^{n+m-1} |\kappa_{j,0}|^2 \tag{9}$$

$$\text{s.t.} \quad l_{int}\Delta\gamma_{a0} \geq \Delta\phi_{min}.$$

The coupling constant, $\kappa_{j,0}$, is the $j$th row and 0th column element of the matrix $(\mathbf{B} + \bar{\mathbf{C}}^{-1}\mathbf{K})$ from the coupled mode evolution Equation (1).

The second optimization is based on Equation (3), the general solution to the coupled mode equations. If we can maximize the projection from the fundamental waveguide mode into a single super-mode, the projection back after the full interaction length should lead to minimal loss. To quantify this metric, we define the projection vectors as follows:

$$\begin{bmatrix} \mathbf{u_0} \ldots \mathbf{u_{n+m-1}} \end{bmatrix} = \mathbf{A^{-1}}. \tag{10}$$

The projection vector for the fundamental mode of waveguide A is $\mathbf{u_0}$. We want $\mathbf{u_0}$ to be a single element vector. The optimization to achieve this can be defined as follows:

$$\min_{w,h,\theta}(|\mathbf{u_0}|^2 - max(u_{0,0}{}^2, \ldots, u_{0,n+m-1}{}^2)) \tag{11}$$

$$\text{s.t.} \quad l_{int}\Delta\gamma_{a0} \geq \Delta\phi_{min}.$$

The minimum value of the optimization is zero. That is achieved when the square magnitude of the vector $\mathbf{u_0}$ is equal to the square magnitude of its largest element, $max(u_{0,0}{}^2, \ldots, u_{0,n+m-1}{}^2))$. This condition is met when the coupled mode system has an exact projection from the fundamental mode of waveguide A to one super-mode of the coupled system.

## 4. Simulation

Using the CMT-guided strategy described above, we simulate all three proposed phase shifter geometries. Our model system consists of an air-clad silicon nitride waveguide with width 1.4 μm, height 280 nm, and refractive index 1.95, carrying 1.55 μm light. This structure satisfies the earlier description of a waveguide A with a TE-like fundamental mode. The three proposed MEMS phase shifter geometries also have a refractive index of 1.95, an interaction length of 100 μm, and starting and final positions of 300 nm and 0 nm as measured from the top of the waveguide to the bottom of the perturbation B, as shown in Figure 1.

The TE-like phase shifter block shares the same dimensions as the waveguide: a width of 1.4 μm and a height of 280 nm. The TM-like phase shifter block is a rotated version of the waveguide, with width 280 nm and height 1.4 μm. Finally, the exact dimension of the trapezoidal geometry was determined by calculating optimal width, height, and sidewall angle from the optimizations described in Equations (9) and (11).

The optimizations for the trapezoidal silicon nitride geometries require a mode solver and an optimization algorithm to solve. We use MIT Photonic Bands (MPB) [28] to simulate the modes and NLopt to implement the DIRECT algorithm for global optimizations [29,30]. The modal simulations were done in a $4\lambda \times 4\lambda$ cell with resolution $\lambda/31$, where $\lambda$ is the wavelength of light set to 1.55 μm. Based on the simulation cell, we set the bounds

for the width as $0 \leq w \leq 2\lambda$ and the bounds for the height as $0 \leq w \leq (4/3)\lambda$. The height is more constrained than the width because of the waveguide height and starting separation. The theta bounds depend on the chosen width and height. The maximum sidewall angle allowed corresponds to the top of the trapezoid not exceeding the maximum width, $\tan \theta \leq (2\lambda - w)/2h$. The minimum sidewall angle corresponds to a triangle, $\tan \theta \geq -w/2h$. Finally, we specify the phase shift as at least $\pi$ because that is the minimum phase range needed in interferometric circuits. The first optimization found a minimum with another TM-like structure of width 172 nm, height 1.432 μm, and 0 rad sidewall angle. The second optimization found a minimum with a small trapezoid with width 172 nm, height 378 nm, and a slight sidewall angle of −78 mrad.

In Figure 2, we show the modal solutions for all four geometries. Visually, the modes of the higher-order system look like a weighted sum of the individual guided modes, which suggests the designs discussed here are well-represented by the coupled mode analysis from earlier.

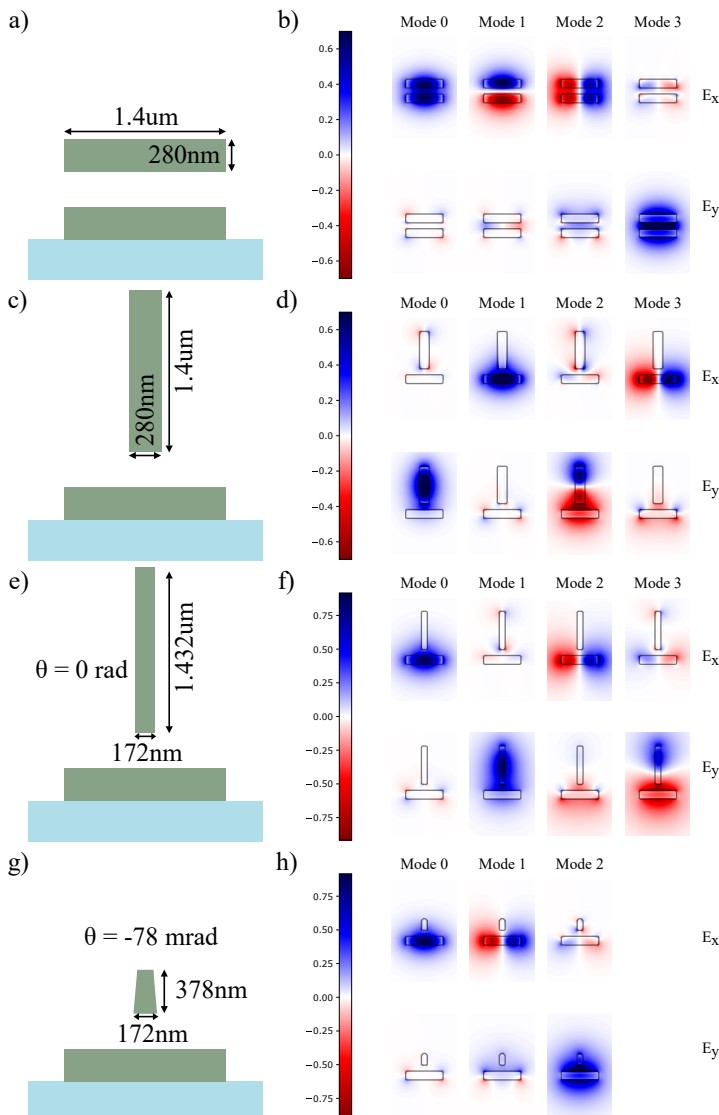

**Figure 2.** (**a**,**b**) A TE-like phase shifter and the transverse electric fields of the first four guided modes of the system. Each column corresponds to a specific mode, and each row corresponds to an electric field component. (**c**,**d**) A TM-like phase shifter with the transverse electric fields of the first four guided modes of the system. (**e**,**f**) The trapezoidal structure found from the first loss optimization condition and the transverse electric fields of its guided modes. (**g**,**h**) The trapezoidal structure found from the second loss optimization condition and the transverse electric fields of its guided modes.

We conducted 3D simulations of all four geometries to sample the transmission and phase shift of the light in the fundamental mode of the waveguide as a function of the perturbation separation. For full transmission simulations, we include a bottom substrate of thermal silicon dioxide of refractive index 1.45.

We simulate the geometries in a generalized scattering matrix formalism using an eigenmode expansion (EME). We discretize the phase shifter along the direction of propagation and use 10 forward and 10 backward modes, including radiating solutions, in each step [31]. This is a computationally efficient method of simulating long structures with high index contrast. When compared to the CMT analysis that guided the design, EME more closely models the behavior of the device, as it includes multiple modes—guiding and radiating—at each cross-section to describe the light propagation. The transmission in this simulation is defined by the the fraction of power in the forward propagating fundamental mode of the waveguide after the phase shifter interaction region.

To verify the EME simulations, we additionally run FDTD simulations using MEEP [32]. We run a full simulation for each change in separation of the phase shifter and waveguide. The step size as the gap approaches zero and the rate of change of the phase increase. Each FDTD simulation used an eigenmode source to excite only the forward propagating fundamental mode of the waveguide. Additionally, an input flux monitor and output flux monitor were placed before and after the phase shifter to capture the transmission.

As shown in Figure 3, the EME and FDTD agree in simulated phase shift response for all four configurations but differ in the transmission as a function of phase. The disparity arises from the finite number of modes in the EME simulation as compared to the continuum of radiation modes simulated in the FDTD case. The EME is more optimistic in the fraction of power coupled back into the waveguide upon termination of the phase shifter. This difference is most notable in the TE-like phase shifter because of the strong coupling into the phase block. Based on the numerical models, there is a trade-off between the phase shift range and the maximum loss. The TE-like phase shifter exhibits a $17\pi$ range, and the TM-like phase shifter exhibits a $2\pi$ range. The two optimal trapezoidal structures achieve the minimum phase shift of $\pi$. The maximum loss is highest with the TE-like structure, which had a starting insertion loss of $-0.92$ dB and a maximum loss of $-2.15$ dB. The TM-like phase shifter had an insertion loss of $-0.02$ dB and a maximum loss of $-0.08$ dB. Finally, the two optimal trapezoidal structures had an insertion loss of $-0.015$ dB and maximum losses of $-0.026$ dB and $-0.022$ dB, respectively. Differences in simulated insertion losses at such small values can be dominated by numerical accuracy, so it is not very meaningful to compare these two optimal solutions against each other. All four simulations are ideal structures that do not account for surface roughness or other fabrication errors.

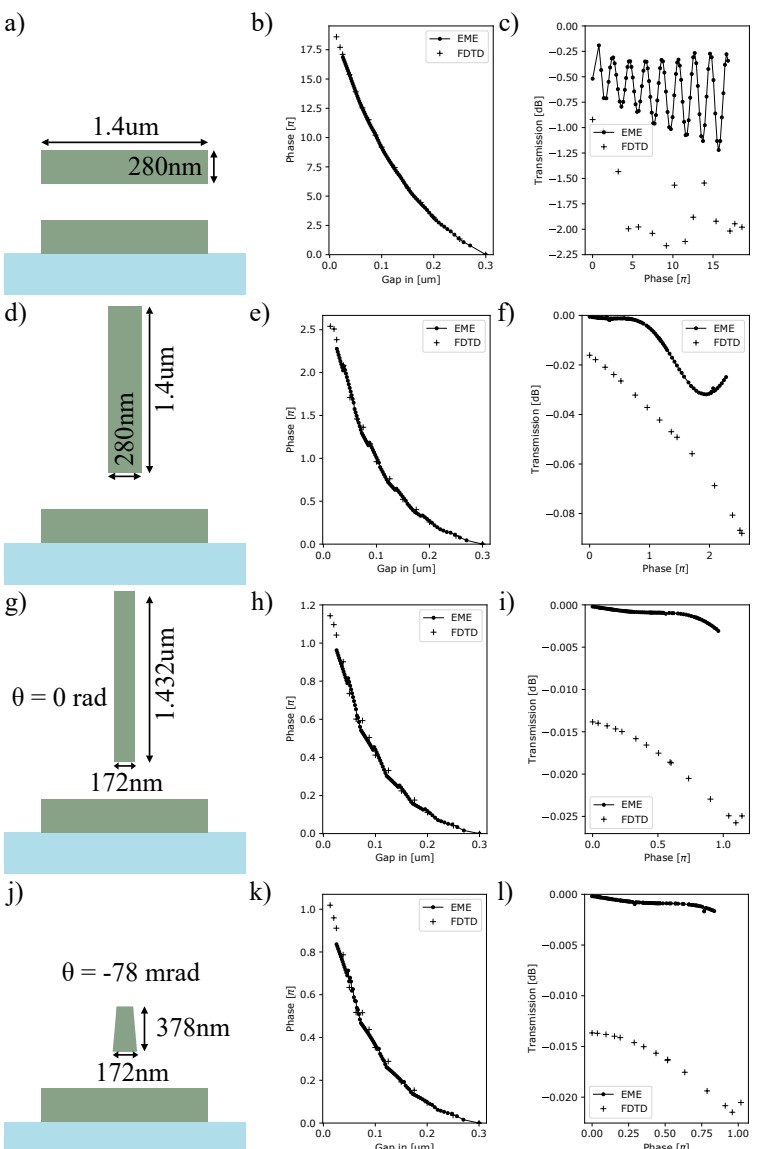

**Figure 3.** (**a**,**d**,**g**,**j**) All four simulated phase shifter geometries. (**b**,**e**,**h**,**k**) The simulated phase shift as a function of separation starting at a gap of 300 nm for each geometry. (**c**,**f**,**i**,**l**) The simulated transmission as a function of phase shift for each geometry.

## 5. Fabrication

For experimental verification, we create a TM-like MEMS phase shifter in silicon nitride following the process presented in [33]. The silicon nitride photonic stack begins with a lightly n-doped <100> silicon handle wafers that undergo wet oxidation at 1000 °C to form 2.16 μm of thermal silicon dioxide (Figure 4a) as the substrate. The waveguide core layer is 280 nm of low-pressure chemical vapor deposition (LPCVD) stoichiometric silicon nitride deposited at 800 °C. This core layer is patterned to form 1.4 μm wide single-mode channel waveguides (Figure 4b). The waveguides are capped with 1.25 μm of plasma-enhanced chemical vapor deposition (PECVD) silicon dioxide deposited at 350 °C (Figure 4c).

The TM-like phase shifter concept requires a high aspect ratio, so we developed a trench-filling approach inspired by the Damascene process [34] used in metal interconnects. As shown in Figure 4d, we open a trench in the oxide cladding that defines the shape of the phase block, partially fill the trench with a conformal sacrificial layer (Figure 4e), and finally completely fill the trench with a conformal silicon nitride layer for the phase block (Figure 4f).

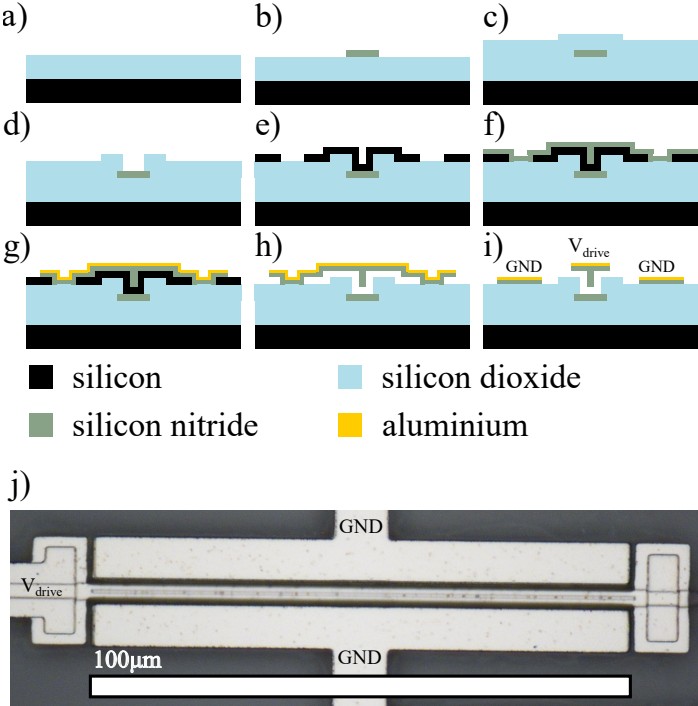

**Figure 4.** (**a**–**i**) Fabrication process for creating a TM-like MEMS phase shifter in silicon nitride. (**j**) Optical micrograph of a 100 μm long MEMS phase shifter with drive voltage, $V_{drive}$, relative to the ground electrodes.

The initial trench width of 1.3 μm is designed to be the target final width—in this case, 300 nm—plus twice the sacrificial layer thickness. Next, 500 nm of a sacrificial layer of LPCVD amorphous silicon is deposited at 580 °C. Amorphous silicon can be removed with a vapor xenon difluoride etch without attacking the photonic or MEMS layers or exerting destructive capillary forces like a wet etchant [33,35]. The high selectivity also eliminates the need for a timed isotropic etch often used in single-layer MEMS photonic designs [24,36,37].

Anchoring points and the bottom electrode are opened by patterning the sacrificial layer (Figure 4e). Next, 200 nm of LPCVD silicon nitride is deposited at 770 °C to fill the trench forming the phase block (Figure 4f). The electrodes and support for the actuator are formed by sputtering 100 nm of aluminum on the mechanical nitride layer and subsequently etching both layers (Figure 4g). Finally, the structure is released with a xenon difluoride isotropic etch of the amorphous silicon (Figure 4h,i). An optical micrograph of the phase shifter is shown in Figure 4j.

## 6. Phase Shifter Performance

The electrostatic micro-bridge that controls the gap between the phase shifting block and the waveguide is actuated by applying a voltage difference, $V_{drive}$, between the center electrode and the grounded side electrodes. To measure the induced phase shift, the MEMS phase shifter was embedded in only one arm of an integrated Mach–Zhender Interferometer (MZI). We used a fiber-coupled tunable laser centered at 1549 nm as the source, grating couplers as the inputs and outputs of the MZI, and an InGaAs photodetector to measure output power. Figure 5a shows the test setup.

Each grating output was collected separately to recover the phase shift as a function of drive voltage. First, the fiber input was aligned to input one, I1, and the cross-state output power was measured at output two, O2. Then, the fiber was moved to input two, I2, and the power was measured out of output one, O1. The cross-state measured powers, I1 to O2 and I2 to O1, were each fit to a unit amplitude cosine function, and the phase was extracted

from that fit. The I1 to O2 and I2 to O1 fits had a starting phase of $-0.23\pi$ and $-0.13\pi$, respectively, with a total phase range of $1.23\pi$ and $1.18\pi$ each.

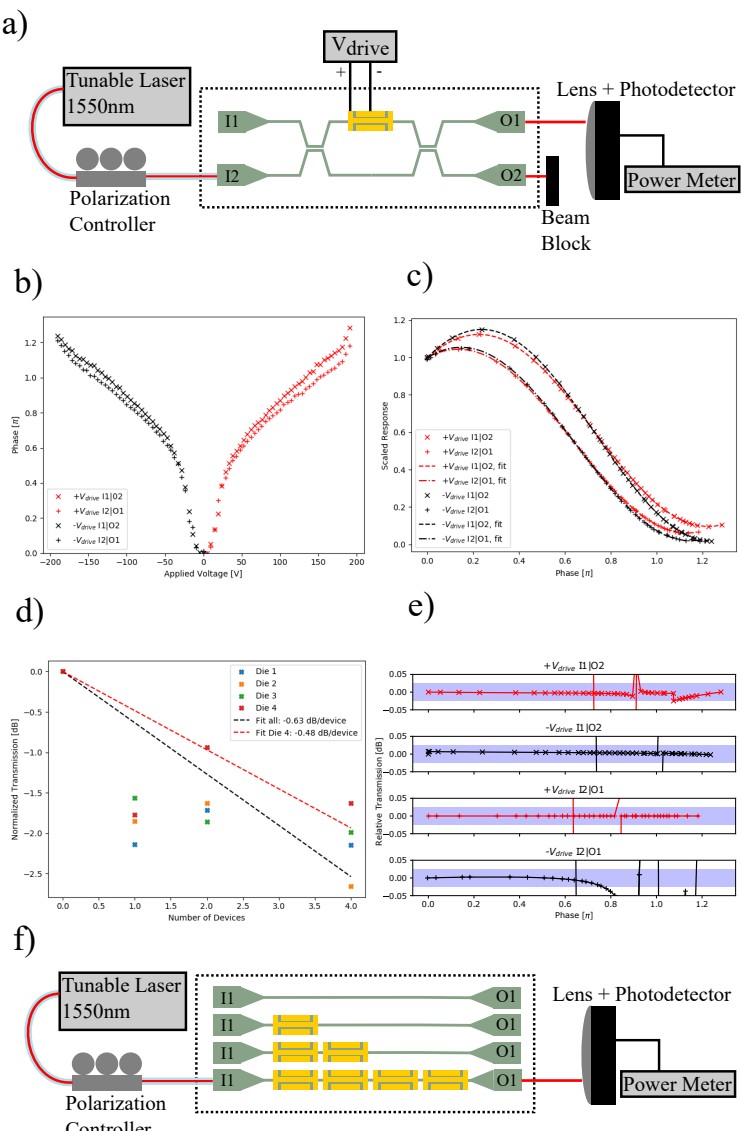

**Figure 5.** (**a**) The measurement setup for the fabricated MEMS phase shifter. A fiber-coupled tunable laser with an external polarization controller is used to couple light into one of the two input grating couplers, I1 or I2. The phase shifter is part of the upper arm of the integrated MZI and driven by a DC voltage source, $V_{drive}$. The outputs of the MZI are measured independently using a beam block between the output grating couplers, O1 and O2, and the collection lens and photodetector. (**b**) The measured phase shift as a function of applied voltage. (**c**) The scaled response of the cross-states of the MZI for positive and negative voltages. The fitted functions used to calculate the transmission of the phase shifter. (**d**) The cutback measurements fitted to find the average insertion loss of $-0.63$ dB and the best insertion loss of $-0.48$ dB. (**e**) The normalized transmission of the fabricated MEMS phase shifter, found from the roots of the quadratic form of the cross-state transmission. The shaded region is $\pm\,0.025$ dB, showing the low additional phase-dependent loss. Uncertainty in the phase in (**b**) results in erroneous transmission variation in the fourth plot of (**e**). (**f**) The measurement setup for the insertion loss is similar to (**a**) but has a single input and output grating coupler for each device and does not need a beam block.

The MEMS phase shifter was actuated in ambient temperature and humidity, which led to charging affecting measurements. To mitigate this effect, we applied alternating positive and negative voltages while flowing dry nitrogen over the structure. As shown in Figure 5b, the fabricated MEMS phase shifter achieves a $1.2\pi$ phase shift over the full $\pm 200$ V range of the high-voltage driver. The phase shift as a function of voltage is approximately quadratic up to $\pm 40$ V. At this voltage, the micro-bridge touches down, so further voltage increase results in reduced increase in phase shift.

To measure the loss over the entire phase shift range, we found the insertion loss of our device with no applied voltage, which is the zero phase shift state, and the relative change in transmission as a function of applied phase shift.

The insertion loss is found by cutback measurements through 0, 1, 2, and 4 phase shifters, as shown in the schematic Figure 5f. The reflections from the input and output grating couplers form Fabry–Perot cavities on top of the measurement. To compensate, each measurement was averaged over four free-spectral ranges from 1548.9 to 1549.1 nm. As shown in Figure 5d, the insertion loss per phase shifter is found by fitting the slope of the measurements. Over the four dies tested, the insertion loss was $-0.63$ dB per device. The best die exhibited an insertion loss as low as $-0.48$ dB per device. Our design focus is on the phase shifter loss and less on average loss due to reflections at transitions. Also, the presence of the aluminum electrode on the phase shifter could be a source of additional loss in the structure. Still, our insertion loss is on par with that of other reported silicon nitride phase shifters that focus on reducing losses at transitions [24].

To calculate the phase-dependent loss, we first fit the cross-state measured response as a function of phase and solve for the transmission from the resulting quadratic form. The measured cross-state response of MZI with a phase shifter on one arm is

$$P_{measured} = P_{in}(\alpha^2 \beta^2)(1 + t^2 + 2t\cos(\Delta\phi + \phi_0)), \tag{12}$$

where $P_{in}$ is the input power coupled into the MZI from the input, $\alpha^2$ and $\beta^2$ are the cross- and straight-through transmission of the directional coupler, and $t^2$ is the transmission of our phase shifter. If the measurement is scaled relative to the zero phase shift state, you can obtain a form independent of the directional coupler characteristics and the input power.

$$\frac{P_{measured}}{P_{measured,0}}(1 + t_0^2 + 2t_0\cos(\phi_0)) = (1 + t^2 + 2t\cos(\Delta\phi + \phi_0)) \tag{13}$$

Using Equation (13), we do a least-squares fit of the scaled measurement to determine $t_0$ and $\phi_0$ for each cross-state measurement described earlier. For this least squares fit, we arrive at the same $\phi_0$ values, $-0.23\pi$ and $-0.13\pi$, as we did for the cosine fit earlier. The resulting fits are plotted in Figure 5c. The difference in the measured transmission of the two cross-states is affected by the fiber realignment between measurements and the charge instability of the MEMS structure when run outside of a packaged environment. The resulting absolute starting phase uncertainty leads to a slightly different normalization—the zero voltage power output—for each curve and thus a different scaled response.

Using the fitted starting points, $t_0$ and $\phi_0$, we solve for the roots of the quadratic form to get $t^2$, relative to $t_0^2$, for each value of $\Delta\phi$. The solutions to the quadratic are plotted in Figure 5e, with a shaded region bounding $\pm 0.025$ dB. The quadratic solution goes through a crossing, and it avoids a crossing that is an artifact of the numerical technique. The fourth plot in Figure 5e suffers from the phase uncertainty of a negative voltage sweep of one cross-state. This phase uncertainty results in an erratic avoided crossing of the quadratic solutions despite the continuity of the measurement and fit in Figure 5c. Combined with the cutback measurement, we demonstrate that our TM-like phase shifter design can achieve an insertion loss of $-0.48$ dB and a phase-dependent loss of less than $-0.05$ dB over a $1.2\pi$ phase shift range.

## 7. Discussion and Conclusions

We have described two approaches based on CMT for the modal design of MEMS phase shifters with low insertion loss. The first approach is to minimize the coupling of power from the waveguide fundamental mode to the modes of the perturbation, leading to a coupled-mode system with analytical solutions. By designing the perturbation to carry an orthogonal fundamental mode, the coupling, and consequently the insertion loss, is reduced. For a TE-like silicon nitride waveguide, we simulated a 100 μm long TM-like silicon nitride MEMS phase shifter with an insertion loss of $-0.08$ dB and a $2\pi$ phase shift. Further optimization yielded a 100 μm long $\pi$ phase shifter with an insertion loss of $-0.026$ dB.

Our second approach is to consider the phase shifter as the complete coupled system and design the perturbation to maximize the projection operation from the fundamental mode of the waveguide to a single super-mode predicted by CMT. Whereas the first approach minimizes the off-diagonal elements of the evolution matrix, the second approach focuses on designing the eigenvectors. The shape of the perturbation is found numerically. Using the same parameterization and constraints as the first approach, a different design optimum composed of a small trapezoidal perturbation is found. This optimized structure has an insertion loss of $-0.022$ dB and a $\pi$ phase shift. These optimizations were limited to three parameters but provide a framework for reaching arbitrarily low insertion loss under prescribed length and phase shift constraints.

To demonstrate the practicality of the first design approach, we fabricated a TM-like MEMS phase shifter design with a process flow that allows for high aspect ratio MEMS to be integrated with the photonics. We tested the TM-like design and achieved $-0.48$ dB insertion loss and less than 0.05 dB variation in transmission over a $\pi$ phase shift range. In terms of total loss, our demonstration is comparable to other MEMS phase shifters in silicon nitride [21,24] that use tapering of the MEMS phase shifters to achieve low loss. Additionally, we explicitly demonstrate a low phase-dependent loss similar to [36].

The proposed modal design of phase shifters is not specific to MEMS. Our CMT analysis is agnostic of the material system, the wavelength of light, or actuation mechanism. We used a MEMS phase shifter based in a silicon nitride process as the model system, but the concepts discussed in this paper can be applied to other integrated photonic platforms to achieve low-loss designs.

**Author Contributions:** Conceptualization, N.S.A. and O.S.; methodology, N.S.A., S.P., R.L.H., P.B., Y.M. and O.S.; software, N.S.A. and S.P.; validation, N.S.A., S.P., R.L.H., P.B., Y.M. and O.S.; formal analysis, N.S.A., S.P. and O.S.; investigation, N.S.A., S.P., R.L.H., P.B., Y.M. and O.S.; resources, O.S.; data curation, N.S.A and O.S.; writing—original draft preparation, N.S.A.; writing—review and editing, N.S.A., S.P., R.L.H., P.B., Y.M. and O.S.; visualization, N.S.A.; supervision, O.S.; project administration, O.S.; funding acquisition, O.S. All authors have read and agreed to the published version of the manuscript.

**Funding:** Office of the Vice Provost for Graduate Education, Stanford University (DARE); Air Force Office of Scientific Research (FA9550-18-1-0186).

**Institutional Review Board Statement:** Not applicable.

**Informed Consent Statement:** Informed consent was obtained from all subjects involved in the study.

**Data Availability Statement:** The original contributions presented in the study are included in the article, further inquiries can be directed to the corresponding authors.

**Acknowledgments:** We thank David A.B. Miller for helpful discussions that motivated potential applications for MEMS optical phase shifters. Part of this work was presented at the Conference on Lasers and Electro-Optics in 2022 (AM2C.1).

**Conflicts of Interest:** The authors declare no conflicts of interest.

## Abbreviations

The following abbreviations are used in this manuscript:

| | |
|---|---|
| MEMS | Micro-Electromechanical Systems |
| CMT | Coupled Mode Theory |
| TE | Transverse Electric |
| MZI | Mach–Zehnder Interferometer |
| InGaAs | Indium Gallium Arsenide |
| LPCVD | Low-Pressure Chemical Vapor Deposition |
| EME | Eigenmode Expansion |
| FDTD | Finite-Difference Time-Domain |
| MBP | MIT Photonic Bands |
| TM | Transverse Magnetic |
| s.t. | such that |

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
