# Peer review of "Coupled Mode Design of Low-Loss Electromechanical Phase Shifters"

_2673-8023, doi:10.3390/micro4020021_

Round 1

Reviewer 1 Report

Comments and Suggestions for Authors

1. Eq. (8) is not clear because Δφmin is not explained in the text.

2. Eq. (9), what does s.t. mean?

3. Figure 5. Caption. Third line…couplers, I1 or I1. Perhaps I1 or I2?

4. When describing Figure 5b (Phase on Applied Voltage), it should be explained how these dependencies were obtained from the experimental data, which obviously represent the Output Power on Applied Voltage dependencies.

5. Figure 5c (Scaled Response on Phase). It is necessary to explain the reason for the difference in the transmission dependences from input I1 to output O2 and input I2 to output O1.

6. Different transmission dependences I1=>O2 and I2=>O1 (Figure 5c) cannot be fitted using Eq. (13) with one set of parameters t0, φ0. Therefore, the possibility of assessing the dependence of the phase shifter transmission on the phase is questionable. Apparently, the model used to obtain the dependence of the MZI transmission on the phase shift should take into account additional parameters describing the real structure of the MZI.

Author Response

We appreciate the thoughtful feedback and have made adjustments to the manuscript to address the feedback. The highlighted text in the most recent revision contains those adjustments. Additionally, we address each point of feedback in this cover letter.

Review #1

  1. Eq. (8) is not clear because Δφmin is not explained in the text.

  • We have adjusted the text around equation 8 to define Δφmin as the minimum phase requirement.

  1. Eq. (9), what does "s.t." mean?

  • We use "s.t." as an abbreviation of "such that". The full phrase has been added to the text and the list of abbreviations.

  1. Figure 5. Caption. Third line…couplers, I1 or I1. Perhaps I1 or I2?

  • Thank you, we have updated the caption to say "...the two input grating couplers, I1 or I2. "

  1. When describing Figure 5b (Phase on Applied Voltage), it should be explained how these dependencies were obtained from the experimental data, which obviously represent the Output Power on Applied Voltage dependencies.

  • We took the output measured power of the cross-states and fit them to a cosine. The fitted cosine starting phases differed slightly for the two cross state measurements were -0.23π and -0.13π with a total phase range of 1.23π and 1.18π, respectively. These were acceptable differences that could be attributed to variation in measurement repeatability as the input alignment is changed between cross state measurements. We have updated the text to reflect these details and provide greater clarity for the reader.
  1. Figure 5c (Scaled Response on Phase). It is necessary to explain the reason for the difference in the transmission dependences from input I1 to output O2 and input I2 to output O1.
  • Similar to the phase fit, the curve fit for the zero-scaled response fit had the same starting phase discrepancy between the cross-phase measurements. The resulting absolute starting phase uncertainty leads to a slightly different normalization, the zero voltage power output, for each curve and thus a different scaled response. The text now reflects this clarification.
  1. Different transmission dependences I1=>O2 and I2=>O1 (Figure 5c) cannot be fitted using Eq. (13) with one set of parameters t0, φ0. Therefore, the possibility of assessing the dependence of the phase shifter transmission on the phase is questionable. Apparently, the model used to obtain the dependence of the MZI transmission on the phase shift should take into account additional parameters describing the real structure of the MZI.
  • A single pair of t0 and phi_0 fits were not sufficient to fit both cross-state measurements. We have updated the text to provide that clarification.
  • A known difference in phi_0 fits as already discussed, required individual fitting. Regardless, the phase range for both measurements is extremely close, so that gives confidence that the fits are measuring the same phase shift effect. With reasonable agreement of the phase shift range between measurements, we believe the trend in transmission over the actuation range can still be fit using the quadratic form in Eq. 13.
  • As shown in figure 5e, the quadratic fit is better at higher transmission and becomes more sensitive closer to the minimum transmission in the cross state. This is reasonable because errors in measuring a low light state become more prevalent at lower absolute powers.
  • The feedback to "take into account additional parameters describing the real structure of the MZI" is unclear. Additional parameters that describe the "real structure of the MZI" would include things like a description of the performance of the directional coupler and grating couplers that were fabricated. These details do not affect the form of the normalized cross-state transmission described in Eq.13, which is why we chose that approach.
  • Many physical parameters can drastically affect the MZI and device performance. These include fabrication details, such as refractive index variation or mechanical creep over time and operation, and test details such as humidity fluctuation which affects charging during operation. Rather than improved modeling of a myriad of parameters, improved environmental controls and test sequence could be used to eliminate the uncertainty in the fits. With further improvements in experimental setup, other measurement techniques could be incorporated to minimize uncertainty in the results.
  • For this demonstration, the experimental work is meant to bolster the main contribution, the design framework. We believe the fabrication and measurements demonstrate the strength of the framework, despite the room for improvements. We have updated the manuscript based on the feedback provided, but we hold our assertion of the value of this framework.

Reviewer 2 Report

Comments and Suggestions for Authors

Manuscript ID: micro-2978118

Title: Coupled Mode Design of Low Loss Electro-Mechanical Phase Shifters

Authors: Nathnael S Abebe et.al.

This manuscript describes a new design of microelectromechanical phase shifters and demonstrates the fabrication and measurement of the designed device. For designing a phase sifter with a minimized loss, the authors analyze the phase shifter based on the coupled mode theory, and obtain the optimized structure to minimize the loss of the phase shifter. Those analyses and the explanations of the loss minimization are valuable and will help readers understand the mechanism of the low-loss electromechanical phase shifter. The authors also carry out 3D simulations using EME and FDTD. Moreover, the authors fabricate one of the investigated devices and achieve -0.48dB insertion loss and less than 0.05dB transmission variation over a π phase shift. Those experimental results are also useful. Therefore, I think this manuscript is published after clarifying the following comments.

Comments

1)      It is unclear how the authors obtained the insertion loss in the simulation. In Figs. 3c), 3f), and 3i), the authors show the phase-dependences of the transmissions. In those figures, we find the phase-dependent loss by the deviation from 0-dB line. In the case of FDTD simulation, there exists a shift of transmissions at phase 0. Is the shift a part of the insertion loss? The authors describe “The maximum loss is highest with the TE-like structure, which had a starting -0.92dB insertion loss. The TM-like phase shifter had an insertion loss of -0.08dB. Finally, the two optimal trapezoidal structures had an insertion loss of -0.026dB and -0.022dB” in the text (lines 221-224). Do the authors obtain those values from Figs. 3c), 3f), and 3i)?

2)      In the actual fabrication, there exists the metal (aluminum) electrode on the movable part of the phase-shifter. Is there any influence on the loss of the phase-shifter?

Author Response

Review #2

1) It is unclear how the authors obtained the insertion loss in the simulation. In Figs. 3c), 3f), and 3i), the authors show the phase-dependences of the transmissions. In those figures, we find the phase-dependent loss by the deviation from 0-dB line. In the case of FDTD simulation, there exists a shift of transmissions at phase 0. Is the shift a part of the insertion loss? The authors describe “The maximum loss is highest with the TE-like structure, which had a starting -0.92dB insertion loss. The TM-like phase shifter had an insertion loss of -0.08dB. Finally, the two optimal trapezoidal structures had an insertion loss of -0.026dB and -0.022dB” in the text (lines 221-224). Do the authors obtain those values from Figs. 3c), 3f), and 3i)?

  • The shift in transmission at phase 0 is the insertion loss, as seen in the simulations. We have corrected the text to state insertion loss when referring to transmission at phase 0, and maximum loss when referring to the lowest transmission over the simulated range. We also included more detailed description of the in-simulation measurement of the transmission in both the EME and FDTD cases. With the corrected text, all the reported values can be seen in the plots in Figure 3.

2) In the actual fabrication, there exists the metal (aluminum) electrode on the movable part of the phase-shifter. Is there any influence on the loss of the phase-shifter?

  • Aluminum in the fabricated device unfortunately can scatter and absorb light based on its proximity to the wave guiding device. This is further exacerbated as the phase shifter is actuated toward the device. For this demonstration, the aluminum sits above 1.25um of top cladding, and when the phase shifter is moved, we do not observe a dramatic change in the total power, so we believe there is a minimal effect.
  • If the aluminum contributes to the insertion loss in the experimental demonstration, we do not attempt to estimate and remove the penalty as part of the reporting because it was needed to form the moving structure. Based on this feedback, we include a statement in the insertion loss discussion drawing attention to the presence of the aluminum.